# Co-PatcheR: Collaborative Software Patching with Component(s)-specific Small Reasoning Models

**Yuheng Tang**[1]    **Hongwei Li**[1]    **Kaijie Zhu**[1]
**Michael Yang**[1]    **Yangruibo Ding**[2]    **Wenbo Guo**[1]

[1]University of California, Santa Barbara    [2]University of California, Los Angeles

{yuhengtang, hongwei, kaijiezhu, yang335, henrygwb}@ucsb.edu
yrbding@cs.ucla.edu

## Abstract

Motivated by the success of general-purpose large language models (LLMs) in software patching, recent works started to train specialized patching models. Most works trained one model to handle the end-to-end patching pipeline (including issue localization, patch generation, and patch validation). However, it is hard for a small model to handle all tasks, as different sub-tasks have different workflows and require different expertise. As such, by using a 70 billion model, SOTA methods can only reach up to 41% resolved rate on SWE-bench-Verified. Motivated by the collaborative nature, we propose Co-PatcheR, the first *collaborative patching system* with small and specialized reasoning models for individual components. Our key technique novelties are the specific task designs and training recipes. First, we train a model for localization and patch generation. Our localization pinpoints the suspicious lines through a two-step procedure, and our generation combines patch generation and critique. We then propose a hybrid patch validation that includes two models for crafting issue-reproducing test cases with and without assertions and judging patch correctness, followed by a majority vote-based patch selection. Through extensive evaluation, we show that Co-PatcheR achieves **46%** resolved rate on SWE-bench-Verified with only $3 \times 14$B models This makes Co-PatcheR the *best patcher with specialized models, requiring the least training resources and the smallest models*. We conduct a comprehensive ablation study to validate our recipes, as well as our choice of training data number, model size, and testing-phase scaling strategy.

## 1   Introduction

Software patching is a time-intensive and cognitively demanding task, especially for large and complex codebases. In the real world, effective patching often requires a combination of *complementary skills*: locating the faulty component, generating plausible fixes, and validating the changes. Recent works leverage general-purpose LLMs [32, 3, 34, 33, 15, 4] to construct patching agents with three components, responsible for localization, generation, and validation, respectively. They demonstrate remarkable performance on SOTA benchmarks (e.g., SWE-bench [22]) and show significant potential to automate patching in the real world. Despite these promising results, concerns about cost efficiency and data privacy further motivate the development of customized patching models. Current approaches train one model for the end-to-end patching pipeline through supervised fine-tuning (SFT) or reinforcement learning. Specifically, early works [54, 29] fine-tune 72B and 32B models through simple supervised data and achieve around 30% resolved rates on SWE-bench-Verified. More recent methods implement rule-based rewards and train reasoning models with reinforcement learning, with SWE-RL [50] achieving the highest resolved rate of 41% on SWE-bench-Verified using a 70B

39th Conference on Neural Information Processing Systems (NeurIPS 2025).

model. However, these monolithic approaches fail to imitate the real-world patching paradigm, where specialized engineers *collaborate by dividing responsibilities according to their expertise*.

Inspired by the collaborative workflow in the realistic software engineering practice [30, 51], we propose Co-PatcheR, the *first patching agent with collaborative small reasoning models designed specifically for different components*. Our key insight for having component(s)-specific models is that different components have different inputs, outputs, and capability requirements. Specifically, localization and generation require a similar capability of interpreting the issue description and understanding the current codebase. Validation, on the other hand, generates testing cases without knowledge of the patches or the codebase. Given the non-trivial differences, it is challenging for one small model to handle all these sub-tasks. Following this intuition, we craft a tailored task design and training recipe for different components, aiming to *minimize the model size while preserving performance*. Specifically, given the similarity between localization and generation, we train a single model (Loc-Gen model) to handle both functions. For localization, we design it as a two-step procedure, where the model first identifies the affected files and then pinpoints the specific lines responsible for the issue. This task decomposition reduces the task complexity and context length, making it more suitable for small models. For patch generation, we train the Loc-Gen model to not only generate patches but also review and refine its own solutions. With this additional self-critical capability, the Loc-Gen model can prevent common errors and generate higher-quality candidates. Finally, we train two models to generate multiple and diverse issue-reproducing test cases (PoC) and judge the patch correctness based on the PoC execution outcomes. The insight here is to provide diverse PoCs for a more sound correctness judgment. Here, Val-assert model and Val-no-assert model generate PoCs with and without assertions, respectively. We use these models together with available functionality tests and a majority vote mechanism to select the final patch. For all three models, we apply model distillation with a novel data construction method to enable their reasoning capabilities. Different from existing distillation models (e.g., S1 [31]), we find that *creating reasoning data with correct answers is critical for our fine-tuned model to achieve high performance*.

Through extensive experiments, we first show that when using only **3×14B** models, Co-PatcheR can achieve a **46%** resolved rate on SWE-bench-Verified with 60 patch candidates. Compared to SWE-RL, Co-PatcheR achieves a high resolved rate with 40% fewer parameters and 88% fewer samples. Besides, Co-PatcheR only needs to run one 14B model at a time, which is much more efficient than SOTA methods during the testing phase. Furthermore, with our specific reasoning data construction method, Co-PatcheR only requires 6K data for training, which is much more efficient than SOTA methods that use at least 30K samples. We then conduct a comprehensive ablation study for each model to validate its task design and training recipe. Finally, we validate the necessity of testing-phase reasoning, our choice of data number and model size, through more ablation studies.

**Contributions.** We propose Co-PatcheR, the first collaborative patching system with component-specific reasoning models. Co-PatcheR is the most data- and parameter-efficient patcher that offers greater effectiveness, efficiency, and modularity than existing patchers with specialized models. Co-PatcheR ranks among the top-10 open-source systems on SWE-bench-Verified, outperforming all patchers with open-source models. We propose specific training recipes for each model and obtain the following new findings that are unique to patching:

- *Using one model for localization and generation performs similarly to using separate models.*
- *Multiple models for PoC generation provide necessary diversity that a single model cannot achieve.*
- *Critique is important for generation, and multi-source data is important for validation.*
- *Simply increasing data or model size is not always helpful; data scale should match model size.*
- *Rejection sampling-based data filtering helps all components; but rationalization does not.*

## 2 Existing Works and Limitations

**LLM-based patching agent.** There are several works on designing a patching agent using general-purpose LLMs [25, 7, 6, 27, 2, 20, 8, 14, 13, 56, 59, 5, 42, 35]. Some agents achieve remarkable performance on the SWE-bench benchmark [22], a benchmark for real-world GitHub issues written in Python. The top-ranked open-source agents are OpenHands [48], Agentless [53], and PatchPilot [24]. Here, Agentless and PatchPilot follow a pre-defined workflow, where PatchPilot introduces a number

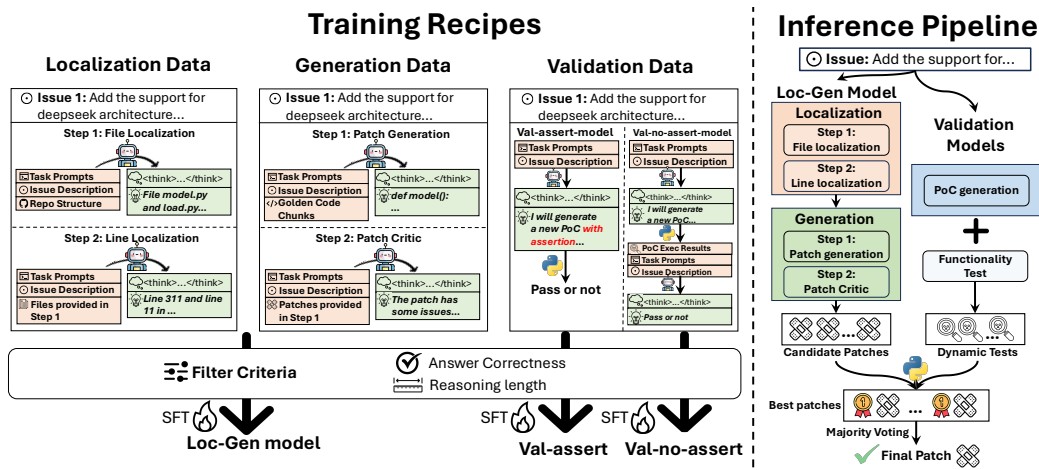

Figure 1: The overall training recipes and inference pipeline of Co-PatcheR. We design one model for localization and generation, where each component has two steps. We further design two models for PoC generation with/without assertions. During inference, we conduct a PoC and functionality testing to select the final patch and conduct a majority vote when dynamic testing has ties.

of optimizations over Agentless. OpenHands, on the other hand, gives more freedom to the LLM to decide its workflow on the fly. OpenHands can have a higher performance but is less stable and more costly than PatchPiolt. We use PatchPiolt as the agent scaffold as it is more cost-efficient.

**Specified models for patching.** There are some early explorations on training customized LLMs for the patching task. At a high level, most methods train *one model* for the end-to-end pipeline, and they use relatively large models. Specifically, SWESynInfer [27] and SWE-Gym [36] train a model with 72 billion (72B) and 32B parameters, respectively, to perform the end-to-end pipeline. Both models are trained with supervised fine-tuning (SFT) without a testing-phase reasoning. Their resolved rate on SWE-bench-Verified is around 30%. SWE-Fixer [54] trains one 7B model for fault localization and a 72B model for patch generation with a resolved rate of 33% on SWE-bench-Verified.

Follow-up works explore training the model with reinforcement learning to enable testing-phase reasoning [58, 29, 50]. SEAlign [58] continues training on SWE-Gym [36] using Direct Preference Optimization [40] to retain preferred solution paths. SoRFT [29] and SWE-RL [50] define rule-based rewards and train the model with policy gradient methods (PPO [43] and GRPO [44]) for both localization and generation. Among these three methods, SWE-RL achieves the highest resolve rate of 41% on SWE-bench-Verified with a 70B model. A concurrent work, SWE-Reasoner [28], on the other hand, applies SFT-based model distillation (from DeepSeek-r1 [15]) to train a 32B reasoning model for the end-to-end pipeline. They further trained two 32B critique models for localization and patch selection. They achieve 46% resolved rate on SWE-bench-Verified with all three models.

**Other code LLMs.** First, there are some coding LLMs for general coding tasks (e.g., LeetCode, Data Science), including Qwen2.5-Coder [19], DeepSeek-Coder [61], WizardCoder [26], CodeLlama [41], and reasoning models (e.g., S* [23] and CYCLE [9]). Second, existing works also explored developing models for debugging [10, 21, 60, 52, 45], testing case generation [1, 18, 38], function and API calling [11, 37], secure code generation [17, 12, 47, 16, 55]. These efforts are orthogonal to our work on training patching-specific models.

## 3 Key Technique

### 3.1 Overview

**Problem setup.** We are given a software repository with one or multiple issues/bugs. Each issue has a simple text description, which may contain additional information such as desired behaviors and vulnerable inputs. Each issue may affect one or more functions in the repository. Our goal is to automatically analyze the issue and generate patches that fix all affected functions while preserving the behaviors of the unaffected functions (which are evaluated by running the hold-out unit test).

**Technical insight.** In this paper, we first argue that designing small and specialized patching models improves the overall efficiency of the patching system, as general-purpose LLMs are way larger. Besides, we do not need the model to process images or video; instead, it must precisely understand the repository, reason about issues, and generate correct patches. Second, we argue that having a single model for the end-to-end patching pipeline may not be the optimal solution given the differences between components and the collaborative nature of software patching. Specifically, both localization and generation need to interpret the issue description and connect it to the target codebase, especially the code chunks responsible for the issue (root cause). Localization needs this capability to scan the entire codebase to pinpoint the root cause, and generation then relies on that information to craft patch candidates. By contrast, test case(unit test) generation during validation demands an even deeper understanding of the issue description, yet it does not need to analyze the full codebase (given that the test cases are typically generated only based on the issue description). Besides, test case generation is typically not aware of patch candidates in order to generate more objective and comprehensive testing. This hypothesis is supported by existing works [50, 54] that trained large models (>70B) using various methods (SFT, offline and online RL) yet achieved relatively low performance (See Section 4). Based on these findings, we propose *to train small but fine-grained models for different components and use them together in the agent system.*

**Technical challenges and key novelties.** The high-level challenge is to *reduce each model to the smallest feasible size without sacrificing too much performance.* More concretely, we need to first design a *data-efficient* training recipe for each model (Challenge ❶). Once we have the models, we also need to decide how to effectively integrate them into the overall agent system (Challenge ❷).

Solve challenge ❶. We propose to train three reasoning models, Loc-Gen model for localization and generation, Val-no-assert model and Val-assert model for PoC test cases generation with and without assertions (Figure 1). First, as shown in Section 4, training reasoning models can achieve a better performance than non-reasoning models even with fewer training samples, making the model training even more data- and cost-efficient. We propose to distill a large reasoning model with supervised fine-tuning. Recent research shows that high-quality distillation data enables training effective small reasoning models for math and coding tasks with limited computational resources [46, 31]. In contrast, training reasoning models with RL requires substantially more samples and computational power, contradicting our efficiency goals. Additionally, without well-designed intermediate process rewards, training based solely on outcome rewards becomes costly and unstable [36]. Second, we train one model for localization and generation as they share similar capabilities. As shown in Figure 1, we divide the localization task into two lower-complexity sub-tasks, generate training data separately, and mix the data for model training. For generations, we integrate "critique training data" where the model reviews its own patches, enabling better reasoning about bug fixing. Third, we design two models for PoC generation to enable more diverse PoC testing. For each model, we train it to (1) generate PoCs that potentially trigger the target issue and (2) evaluate patch correctness based on issue descriptions and PoC execution outcomes.

Solve challenge ❷. Figure 1 shows our proposed agent workflow, which is inspired by the efficient designs of PatchPilot [24]. Our localization first identifies the files and then the lines in the pinpoint files for locating potential root causes. The generation component then generates multiple patch candidates. Finally, we use our two PoC generation models for patch correctness testing, followed by a model-free functionality test that runs patches against public functionality tests. We rank patches based on dynamic testing results (Num. of passed PoC and functionality tests) to identify the highest-scoring candidates. When multiple patches achieve the same highest score, we apply majority voting based on normalization [53] to select the final patch.

## 3.2 Training Recipe for Loc-Gen model

Our training recipe has four key components: training issue selection, training task construction, reasoning data generation, and filtering. Issue selection and data filtering are common across all components, while task construction and reasoning data generation are tailored to each model.

### 3.2.1 Issue Selection

We select training issues and the corresponding codebases from the SWE-bench training set and SWE-Gym dataset, which contains *different repositories from our testing set*. Our selection criteria focus on two key factors: First, we prioritize *diversity* by selecting issues from different repositories

across different time periods to improve model generalization. Second, we include issues with *varying difficulty levels*, following recent work in S1 [31] showing that challenging cases improve reasoning abilities with limited training data. We quantify the difficulty using the number of files changed in the fix. For example, simple issues require changing only one file, while difficult issues require changes to five or more files. As shown in Section 4, in general, the performance of our models first increases as the training data grows and then remains at a similar level. Guided by this trend, we select only *2,000* training issues, which is significantly fewer than existing RL-based methods, e.g., 11M in SWE-RL [50]. To avoid data leakage, we check the selected issues, making sure they come from different codebases from the testing ones and do not have overlapping functions.

### 3.2.2 Task Construction and Reasoning Data Generation

**Localization.** The *key challenge* for localization is to efficiently identify root causes while keeping the task manageable for a small model. To achieve this, rather than training the model to directly identify root causes from the entire repository, we decompose localization into two sequential sub-tasks: file localization and line localization. File localization identifies issue-related files based on the file structure, while line localization pinpoints the specific code chunks within selected files. This decomposition creates simpler sub-tasks with shorter context, better suited for small models. It also provides explicit guidance to the localization process. Specifically, for file localization, we provide the issue description and codebase file structure as input. For line localization, we input the issue description and the code from each selected file (splitting files that exceed the model's context limit) (see Appendix E for the prompts). Based on this task design, we use Claude-3.7-Sonnet [4] to generate distillation data with a reasoning chain. The reasons for choosing Claude-3.7-Sonnet over other reasoning models are that some models (e.g., OpenAI models) do not provide their reasoning chains. Among the two main open-source models, DeepSeek-R1 [15] tends to give overly long reasoning chains, making it easy for our model to learn noisy steps. QwQ [39], meanwhile, performs poorly on patching-related tasks.

**Generation.** The *key novelty* of our generation model is to combine the patch critique with the patch generation. For patch generation, we design the input as the issue description and the identified root causes, and the output as a patch candidate. For the patch critique, our input is the issue description and a patch candidate, and the designed output is the review of the candidate as well as a correct answer if it is wrong [49]. We still use Claude-3.7-Sonnet to generate the reasoning data for these two sub-tasks. Having the critique data can guide the model to learns not only to produce an answer but also to diagnose and refine existing ones, thereby acquiring stronger reasoning skills during training. Such a process could further deepen the model's understanding of the target issues and potentially yield higher-quality patches. Appendix E specifies our input prompts.

### 3.2.3 Reasoning Data Filtering

After generating the data, we apply filters based on two aspects: final answer correctness and reasoning length. First, based on our empirical observation, we conduct a *rejection sampling* to filter out the training samples that lead to wrong answers, as training with these noisy samples will jeopardize our model performance. This is *a unique recipe for patching*, as it does not align with existing work, Sky-T1 [46] and S1 [31], where they state that data with wrong answers is still useful for models to learn the reasoning structure in math problems. We believe the difference stems from the specialized nature of patching, where the related tasks are not frequently encountered during pre-training. As such, a small model needs access to correct answers to learn the correct knowledge. For general maths problems, however, the model has likely seen enough examples in pre-training and the model can tolerate occasional wrong answers. Here, reasoning data mainly teaches it to perform reasoning following a certain structure. Second, we filter out samples with excessively long reasoning chains, as these kind of long reasoning does not offer too much benefit even on general-purpose LLMs (Appendix C). A deep inspection of the reasoning chain shows that the model tends to overthink and repeat reasoning paths. Such data can even cause model collapse and jeopardize training efficiency.

### 3.3 Training Recipe for Val-assert model and Val-no-assert model

**Rationale for having two models.** The high-level goal of validation is to decide whether a candidate patch fixes the issue (patch correctness) and whether it affects other benign functions (functionality correctness). For functionality correctness, we can retrieve the public unit test cases from each project

Table 1: Co-PatcheR vs. baselines on SWE-bench-Verified. N/A means not available.

| Model | Agent scaffold | Training | Reasoning | Resolved (%) | Training data | # candidates |
|---|---|---|---|---|---|---|
| *General Commercial LLMs* | | | | | | |
| Claude-3.5-Sonnet | Agentless | - | ✗ | 50.80 | N/A | 40 |
| Claude-3.5-Sonnet | OpenHands | - | ✗ | 53.00 | N/A | N/A |
| Claude-3.5-Sonnet | PatchPilot | - | ✗ | 53.60 | N/A | 12 |
| Claude-3.7-Sonnet | OpenHands | - | ✓ | 60.60 | N/A | N/A |
| *Specialized Open-source Models* | | | | | | |
| Lingma-SWE-GPT-72B | SWE-SynInfer | SFT | ✗ | 30.20 | 90K | N/A |
| SoRFT-32B | Agentless | SFT | ✗ | 30.80 | 30K | N/A |
| SWE-Fixer-72B | SWE-Fixer | SFT | ✗ | 32.80 | 110K | 1 |
| SWE-RL-70B | Agentless-mini | RL | ✓ | 41.00 | 11M | 400 |
| Co-PatcheR (3×14B) | Agentless-mini | SFT | ✓ | 45.20 | 6K | 60 |
| **Co-PatcheR (3×14B)** | **PatchPilot** | **SFT** | ✓ | **46.00** | **6K** | **60** |

and run dynamic testing against our patches. The *key challenge* is to design the patch-correctness validation. To enable dynamic testing, we propose to train two validation models to generate PoC test inputs and make a patch correctness judgment. The definition details about two kind of test case format are explained in Appendix B. The insights for having two models are as follows. First, existing patching agents [24, 53] have two ways of generating PoC tests: with or without assertions. Here, assertions mean specific assertion instructions in the PoC test that judge whether the PoC execution triggers the target issue. The test cases with and without assertions typically cover different program paths to the root cause site. To enable more comprehensive and sound PoC tests, we aim to generate PoCs in both styles. As such, we train two different models, one for each style. As shown in Appendix C.3, we also train one model to generate both types of PoCs with different input prompts and special tokens. However, the model cannot give PoCs with enough diversity, even with a high temperature during testing.

**Training recipe.** Here, we use the same set of training issues as the Loc-Gen model. We design two types of input prompts to instruct the teacher model to generate PoCs with and without assertions (Appendix E). Both input prompts contain the issue description and a format instruction (with/without assertions). Different from Loc-Gen model, we use two teacher models, Claude-3.7-Sonnet and o4-mini, to collect the reasoning data. The goal here is again to increase the PoC diversity and thus path coverage to the root causes. For Val-no-assert model, we further gather judgment data, where the input is the issue description, the current patch, and its PoC execution outcomes, and the output is whether the patch fixes the issue. We train Val-no-assert model to generate the PoCs and judge the patch correctness at the same time. For Val-assert model, we only train it to generate the PoCs, as the PoC correctness can be decided by assertions. As shown in Figure 1, we run dynamic testing with PoC and functionality tests, and conduct a majority vote to select the final patch when dynamic testing has ties.

## 4 Evaluation

### 4.1 Co-PatcheR vs. Baselines on SWE-bench

**Setup and design.** We adopt the Qwen-2.5-Coder-14B model [19] as our base model for all three components. Compared to more recent models, Qwen-2.5-Coder-14B has the knowledge cut of March 2024, which is prior to the SWE-bench benchmark (published in May 2024). It is less likely to be trained specifically for the SWE-bench data. As introduced in Section 3.2.1, we select 2K training issues from the SWE-bench training set and the SWE-Gym [36] dataset and conduct filtering to avoid data leakage. After training our three customized models, we integrate them into our end-to-end pipeline (Figure 1) and evaluate our system (Co-PatcheR) on the SWE-bench-Verified dataset. The specific training hyper-parameters are shown in Appendix A. During the inference, for every issue, we generate 5 root causes from localization, 60 candidate patches, and 4 PoCs, using them to get one final patch. We compare Co-PatcheR with SOTA agents built on commercial LLMs and those with open-source models. We report the resolved rate of these agents' final patch (*best@k*), as well as their number of patch candidates $k$ (if available). For open-source models, we also compare Co-PatcheR with them in training data and model size. Note that a recently released concurrent arXiv work (SWEReasoner [28]) claims a 46% resolved rate with 3 × 32B models. We achieve the same resolved rate with over 50% smaller models.

**Results.** Table 1 shows the comparison between Co-PatcheR and the baseline methods. As we can first observe from the table, most existing specialized models have a large performance gap from the agents with commercial models. SWE-RL archives the highest resolved rate with a 70B model with 110M training data and 500 candidate patches. In comparison, Co-PatcheR sets a new open-source record with a resolved rate of $46.00\%$ using only $3 \times 14B$ models trained with 6K training data. This result validates the advantage of having component-specific models over one end-to-end model when patching with small models. It also demonstrates the effectiveness of our issue selection and reasoning data generation and filtering methods, which significantly improve Co-PatcheR's data efficiency. Besides, the resolved rate of Co-PatcheR ranks among the top-10 open-source tools on SWE-bench-Verified, beating many agents with commercial models. The result shows the importance of having specialized models for software patching. Finally, Table 1 shows the advantages of reasoning models for both general and specialized LLMs. For example, OpenHands has a 7% improvement when using Claude-3.7-Sonnet (reasoning model) compared to Claude-3.5-Sonnet (non-reasoning model). At the same time, Co-PatcheR and SWE-RL also have significant advantages over other baselines with non-reasoning models.

## 4.2 Effectiveness and Ablation Study of Each Component

### 4.2.1 Localization

**Design.** We evaluate our localization component against three commercial LLMs (GPT-4o, Claude-3.7-Sonnet, o4-mini) on SWE-bench-Verified, measuring both file-level and line-level localization accuracy. To isolate the effect of our data filtering strategy, we also train a comparison model (Loc-NoFilter) with unfiltered data containing both correct and wrong answers, using the same 2K data size for fair comparison. We also compare against our base model (Qwen-2.5-Coder-14B) to demonstrate the impact of our specialized training. For all models, we select Top@5 files and report whether the correct answer appears in the root causes identified from these files. For issues affecting

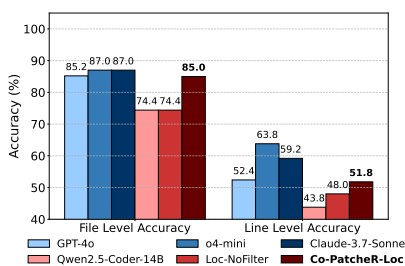

Figure 2: The top@5 file-level and line-level accuracy for localization.

multiple files or lines, we enforce strict evaluation criteria, counting a localization as correct only when it identifies the complete set of affected files and lines. Note that we do not consider training a model to directly identify vulnerable lines from the entire repository, as it will exceed the model's context limit.

**Results.** As shown in Figure 2, SOTA commercial reasoning models o4-mini and Claude-3.7-Sonnet achieve the highest performance on both file and line levels, marginally outperforming Co-PatcheR-Loc. However, Co-PatcheR-Loc achieves comparable performance to GPT-4o, demonstrating the advantage of specialized reasoning models over general non-reasoning models. These results support our claim that specialized models with proper testing-phase scaling can compete with much larger commercial LLMs on specialized tasks. The substantial performance gap between Co-PatcheR-Loc and both Qwen-2.5-Coder-14B and Loc-NoFilter validates the effectiveness of our training recipe, particularly our reasoning data filtering approach.

### 4.2.2 Generation

**Design.** Following the experiment design for localization model, we evaluate our generation component against commercial LLMs (GPT-4o, o4-mini, Claude-3.7-Sonnet) and our base model (Qwen-2.5-Coder-14B). To isolate the contributions of our training innovations, we test two additional variants: Gen-Base (using unfiltered reasoning data without critique training) and Gen-NoFilter (adding critique data but without data filtering) to verify the effectiveness of both data filtering and critique training techniques. For a fair comparison and to focus specifically on patch generation capabilities, we use GPT-4o localization results as consistent input across all models, evaluating

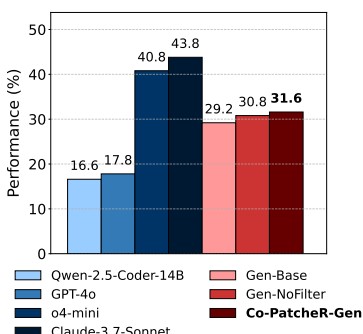

Figure 3: The pass@1 resolved rate for generation.

the performance using the *pass@1* metric, which evaluates the successful issue resolution with only one generated patch.

**Results.** Figure 3 shows the pass@1 performance across models, with results consistent with our localization experiments: o4-mini and Claude-3.7-Sonnet outperform Co-PatcheR-Gen in single-patch performance. However, as demonstrated in Figure 5b, if effectively leveraging our testing-phase scaling approach, Co-PatcheR-Gen achieves comparable performance to these much larger models when generating only 4 more patch candidates. Furthermore, the performance advantage of Co-PatcheR-Gen over both Gen-Base and Gen-NoFilter validates our novel designs: critique training and data filtering substantially improve patch quality. We note that GPT-4o's unexpectedly low performance stems primarily from formatting issues, as it frequently generated syntactically invalid patches that did not follow our required format specification.

### 4.2.3 Validation

**Design.** We conduct the ablation study for both the PoC generation model and the validation workflow, respectively. For the PoC generation model, we compared four variants: (1) Val-no-assert-Base model, trained with reasoning data from Claude-3.7-Sonnet without filtering, (2) Val-no-assert-NoFilter, trained using both Claude-3.7-Sonnet and o4-mini reasoning data without filtering; (3) Val-no-assert model only, and (4) Co-PatcheR-Val: Val-assert model +Val-no-assert model. We integrated each model into our validation pipeline and measured their resolved rates on an identical set of 20 patch candidates produced by our generation component. A higher resolved rate indicates more effective validation.

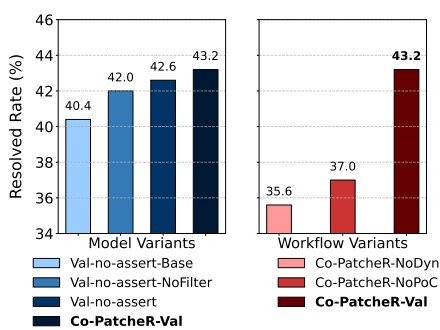

Figure 4: The resolved rate for different validation models and validation workflow.

To evaluate our validation workflow design, we test three strategies with the same Val-assert model +Val-no-assert model: (1) Co-PatcheR-Val, which applies the whole workflow, (2) Co-PatcheR-NoPoC, which omits PoC testing and relies solely on functionality tests and majority voting, and (3) Co-PatcheR-NoDyn, which applies majority voting directly to patch candidates without any dynamic testing. Each workflow also processed the same set of 20 patch candidates for fair comparison.

**Results.** Figure 4 presents the comparative performance of different validation models and workflows. First, using two models performs better than only having Val-no-assert model, confirming the better PoC diversity. Second, Val-no-assert model outperforms Val-no-assert model-NoFilter, confirming the generalizable effectiveness of our data filtering strategy across all components. Comparing Val-no-assert model-NoFilter with Val-no-assert model-Base further justifies the necessity of having diverse PoCs in training data, which guide our model to learn to generate multiple PoCs for the same issue. The results in Figure 4 further show the necessity of having both PoC tests and function tests during validation. In Appendix D, we show that when having 60 patch candidates, majority vote is more effective than the outcome reward model used in SOTA agents [48] and even Claude-3.7-Sonnet. As such, we stick to the majority vote as the final patch selection.

### 4.3 More Ablation Study and Sensitivity Test

We use our generation model to conduct the ablation study on data size, model size, and testing-phase scaling strategy. The results of the other two components are consistent (Appendix C).

**Data size.** We randomly sample a subset of 500 and 1K cases from our current 2K training set and train two models using our proposed recipe. We report the Pass@1 performance of these models in Figure 5a. The result shows that the performance increases more as the parameters grow from 500 to 1K than from 1K to 2K. As shown in Appendix C.2, the model performance for localization no longer increases as we further increases the training data to 5K. As such, we select 2K as our final training data size. The findings show that, for small models, continually adding more data does not guarantee better performance (given the risk of overfitting). We further train a non-reasoning model for patch generation (SFT with ground truth patches). Our result shows that a non-reasoning model trained

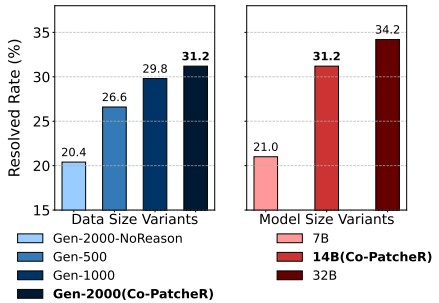

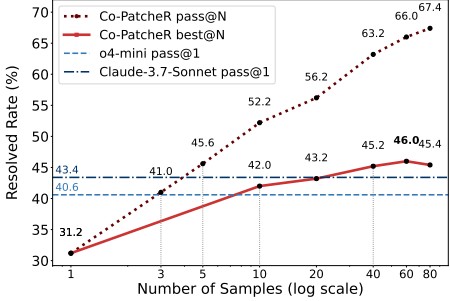

(a) The pass@1 resolved rate of different training data/model size for validation.

(b) The resolved rate for # of patch candidates.

Figure 5: More ablation studies on the generation component.

with 2K training data performs even worse than our reasoning model trained with 500 samples. It further shows that the reasoning model is more data-efficient.

**Model size.** We change our base model to Qwen-2.5-Coder-7B and Qwen-2.5-Coder-32B (same model family with different sizes) and retrain our patch generation model with the same training data. The Pass@1 results in Figure 5a show that a larger model indeed improves the performance. However, considering that the improvement of the 32B model over the 14B model is not significant, we still choose the smaller one.

**Testing-phase scaling.** We test two scaling performances. We fix the output context limit and ask the model to generate K=1, 10, 20, 40, 60, and 80 candidate patches. For each setting, we 1) compare the *pass@K* resolved rate (whether correct patch is in the generated candidates) to obtain the upper bound of patch generation; 2) run our validation to select the final patch (*best@K*) to assess the upper bound of Co-PatcheR. As shown in Figure 5b, increasing the sample numbers can prompt the model to generate more diverse patches, which increases the chances of hitting the correct one. This validates our arguments that small models with many samples can reach a similar performance to large models with fewer samples (without requiring significantly more computing, as the models are much smaller). Increasing sample numbers can also help the system as a whole; however, having too many samples will add a burden to validation and may jeopardize the validation accuracy.

## 5 Discussion

**Rationalization does not always help.** In patch-generation training data collection, we tried a rationalization scheme [57]: We provide the teacher model with the ground-truth patch and force it to generate a reasoning without mentioning the ground truth patch. When context is insufficient, the model invents latent details (e.g., suggesting a likely `some_function` that is not in the context), causing the student model to learn hallucinated patterns. Of ten instances that Co-PatcheR originally solved but fail after fine-tuning with the reasoning data, six fail due to hallucinated identifiers. Thus, rationalization can degrade patch-generation performance.

**Component specific models vs. one model.** In this paper, we argue that to minimize the model sizes, we need to train models specific to individual components. However, a counterargument for promoting one end-to-end model could be that all three tasks work on the same codebase, and the knowledge about the codebase can be shared across tasks. Although we acknowledge the validity of this argument, we do not take this route as we aim to push the limits for small models, and existing works following this methodology show limited performance. Future works can explore the efficient training methods and proper model sizes for such a unified model.

**Limitations and future works.** First, designing specific and effective reward functions requires non-trivial effort. We defer to future work to explore effective RL methods to continue training our current models and see if the performance can be further improved. Second, given our focus on the model side, the current patching leverages a simplified agent scaffold without complex tool calls. We will further enrich the agent with more tool calls and train specified models for tool call planning. Third, with large samples, our localization and generation components can reach the performance of SOTA commercial models. Future works will explore how to design more effective validations to pinpoint the correct patch from many candidates.

# 6 Conclusion

In this paper, we propose Co-PatcheR, a novel software patcher with collaborative small reasoning models. We develop a unique training recipe for models in three major components of the patching pipeline (localization, generation, and validation). Our results show that Co-PatcheR achieves one of the highest resolved rate among patchers with customized models with the smallest models, outperforming many commercial LLM-based patchers. Our ablation studies validate the key designs in our proposed recipes as well as our choices on data, model, and testing-phase scaling strategy.

## Acknowledgements

This work was supported in part by ARL Grant W911NF-23-2-0137. We gratefully acknowledge the support of FAR AI, OpenAI, Anthropic, Google and Berkeley RDI.

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

# A  Training Details and Inference Efficiency Metric

**Hyper-parameters.** We train all three customized models (Loc-Gen, Val-no-assert, Val-assert) using the same settings. Table 2 summarizes the key hyper-parameters.

**Inference efficiency metric.** we comprehensively evaluated the efficiency of Co-PatcheR, as shown in the table 3. We log four kinds of metrics for an average of every issue: end-to-end wall-clock time for one instance under our testing setup(5 root causes from the localization model and 4 PoCs from the validation model), tokens generated across all model calls (completion), latency for average one-shot model inference, and peak GPU memory. These metrics provide the efficiency baseline for Co-PatcheR on two NVIDIA L40S (48 GB) cards with a VLLM deployment framework. The throughput can be further improved through parallel processing of multiple instances.

Table 2: Training hyper-parameters for the Qwen-2.5-Coder-14B model

| Hyperparameter | Value |
|---|---|
| Peak learning rate | $1\times10^{-5}$ |
| Warmup ratio | 0.10 of total steps |
| LR scheduler | Cosine decay (with 10% linear warmup) |
| Batch size (per GPU) | 1 (effective batch size = $1\times12$ accumulations) |
| Weight decay | $1\times10^{-5}$ |
| Number of training epochs | 3.0 |
| Maximum sequence length | 32768 tokens |

Table 3: Inference efficiency metric comparison for Co-PatcheR and SWE-Fixer

| Metric(per issue) | Co-PatcheR | SWE-Fixer |
|---|---|---|
| End-to-end wall-clock time | 334.17 s | 123.25 s |
| Model-only latency | 44.01 s | 103.08 s |
| Tokens generated | 5.83 k | 2.51 k |
| Peak GPU memory (per GPU) | 44.64 GB | 45.23 GB |
| Resolved Rate (%) | 46.00 | 32.80 |

# B  Explanation about Validation Models

We design two format of validation test case generation, the assertion tests independently determine whether the bug is triggered, whereas non-assertion tests only record the output of the PoC and require an LLM to judge the output after execution.

**Assertion tests** follow the common unit test style, which would provide an explicit oracle to the target issue—for example, assert sort([2,1]) == [1,2]. When such a test fails, we know that the bug is highly likely to still exist. However, the downside is that writing a correct oracle requires the model to infer an exact post-condition for the issue, which can be difficult.

**Non-assertion tests** relax this requirement by performing a general PoC script. These are similar to differential testing, where we compare execution outputs before and after applying the patch to determine fix effectiveness. They are generated to a PoC according to the issue description, being expected to reproduce the same observable symptoms. In the downstream workflow, we would ask LLM to make a judgment according to the original issue description and the output of non-assertion tests. And another benefit from non-assertion tests rather than assertion tests is that non-assertion tests can capture the new bug introduced by the candidate patch by observing and analyzing actual output behavior.

We keep both kinds of tests for two reasons: **Coverage** – Some bugs have well-defined expected values or exceptions that can be explicitly tested with assertions, while others manifest as subtle behavioral changes or output variations that are better captured through execution observation rather than rigid assertion checks. Using both styles ensures we can validate a wider range of issues. **Regression** – A patch might solve the original failure but introduce a new one. Because non-assertion

Table 4: Effect of reasoning length on patch generation (`pass@1`) on the SWE-bench–Verified dataset.

| Setting | Max reasoning tokens | pass@1 (%) |
|---|---|---|
| No-reason | 0 | 40.6 |
| Short-reason | 2K | 43.8 |
| Long-reason | 8K | 44.2 |

tests observe the output of the program, they can detect these new bugs even though the old assertion still passes. The whole patching system can utilize this information for further patch refinement.

## C  Ablation Study

### C.1  Whether Long Reasoning Help

**Design.** To compare the impact of reasoning length in the patching task, we ran an ablation study on the SWE-bench–Verified dataset using Claude-3.7-Sonnet. For a fair comparison and to focus specifically on patch generation capabilities, we use GPT-4o localization results as consistent input as 4.2 to patch generation. We then varify the maximum reasoning output length, setting it to 0 tokens("no-reason"), 2K tokens ("short-reason") and 8K tokens ("long-reason"), and evaluate performance with the `pass@1` metric, which counts issues successfully resolved by the first generated patch.

**Results.** The `pass@1` results were 43.8% for the short reasoning setting(2K) and 44.2% for the long reasoning setting(8K), while 40.6% for the no-reasoning settings. These results show that considering the random deviation of the model inference, extending the reasoning budget beyond 2K tokens yields no appreciable gain in our experiments.

### C.2  Localization

We use our localization model to conduct the ablation study on data size, model size, and testing-phase scaling strategy.

**Data size.** We randomly sample a subset of 500, 1K, and 2K cases from our 5K training set and train four models in total using our recipe. We report both the file-level and line-level localization performance of these models in Figure 6a. The result shows that the performance keeps increasing from 500 to 1K and from 1K to 2K, and the model performance for localization no longer increases as we further increase the training data to 5K. As such, we select 2K as our final training data size. We further train a non-reasoning model for localization (SFT with ground truth files and lines). Our result shows that a non-reasoning model trained with 2K training data performs even worse than our reasoning model trained with 500 samples. It further shows that the reasoning model is more data-efficient.

**Model size.** We change our base model to Qwen-2.5-Coder-7B and Qwen-2.5-Coder-32B (same model family with different sizes) and retrain our model with the same localization training data. The localization results in Figure 6b show that a larger model indeed improves the performance. However, considering that the improvement of the 32B model over the 14B model is not significant, we still choose the smaller one.

**Testing-phase scaling.** We test two scaling performances here. The generation setting is kept fixed, whereas the localization model predicts N=1, 2, 4, 6, 8, 10 times for each issue, including both line-level and file-level. For each setting, we 1) fix the top 5 files as file-level results and merge the line-level localization outputs as the line-level results accuracy. 2) generate 20 patches with the generation model with the different localization results as input, then record the *pass@20* resolved rate(the percentage of which at least one of the 20 candidate patches fixes the issue)—this serves as the upper bound of the localization module. To clarify the relationship between our localization and patching metrics, we first note that line-level localization accuracy is defined by whether the model's returned lines fully cover all modified lines in the golden patch—only perfect coverage counts as "correct". In other words, our line-level metric is very strict: it says a location is "correct" only if the predicted lines match the golden patch line for line. But an issue can be fixed without copying the golden patch exactly—any edit that makes the tests pass is accepted. So a patch can solve the issue

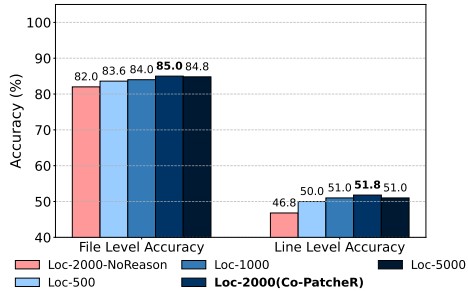

(a) The top@5 file- and line-level accuracy versus training data size.

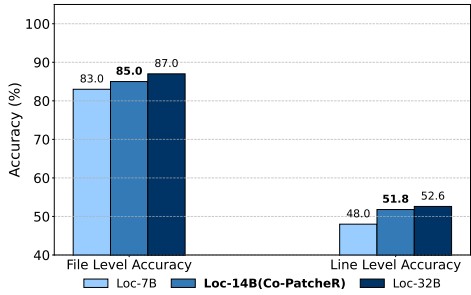

(b) The top@5 file- and line-level accuracy versus model size.

Figure 6: Ablation studies on data size (A) and model size (B) for localization.

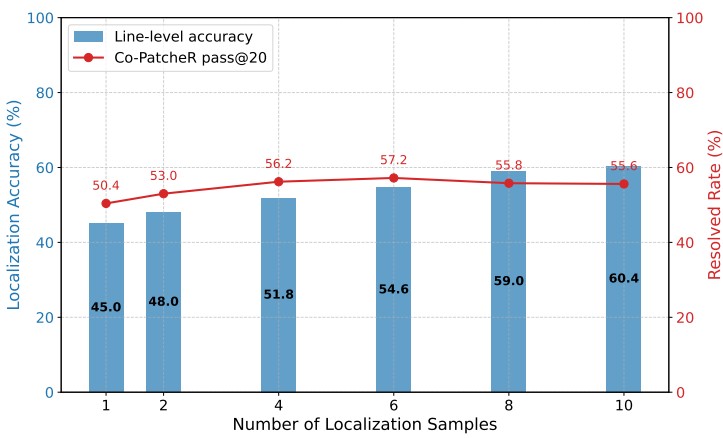

Figure 7: Resolved rate and line-level accuracy for # of localization samples.

even when its edited lines do not align with the golden patch, which is why the *pass@20* success rate can be higher than the measured line-level accuracy. As illustrated in Figure 7, supplying more localization samples broadens the search space, boosting the chance that the correct file appears in the pass@20 candidates. This confirms our intuition that "wider" localization (more samples) can offset a smaller model size just as "wider" generation does, without incurring the cost of a larger backbone. However, pushing the number $N$ too high again shifts the burden to generation and can confuse the generation by a large number of false positives of code chunks, resulting in a decrease in performance.

### C.3 Validation

**Combine training.** In the original pipeline, we train two separate PoC-generation models— Val-assert model and Val-no-assert model —and use them to create four PoCs (two of each type) for validating the candidate patches produced by the generator. To test whether a single, mixed validator can do better, we merge the two training sets, train one unified model, and invoke it with two prompt templates (assert / no-assert) to generate the same four PoCs. Everything else is held constant: the localization and generation results, the dynamic tester, and the ranking heuristic. We then compare the *best@20* resolved rate after validation has selected the final patch as our result. Table 5 indicates that keeping the PoC generation model separate still yields the higher accuracy; combining the data into a single model does not translate into better PoC generation due to a lack of diversity.

Table 5: Split vs. unified PoC-generation models (*best@20* on SWE-bench–Verified).

| Training scheme | PoC prompts | best@20 (%) |
|---|---|---|
| Split models (Co-PatcheR) | 2 assert + 2 no-assert | 43.20 |
| Unified model | 2 assert + 2 no-assert | 42.60 |

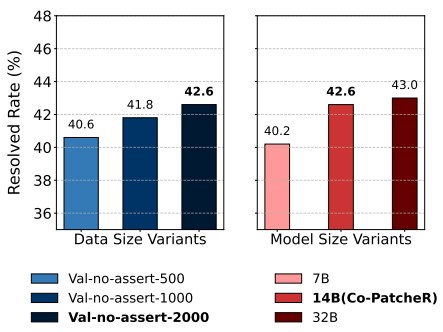
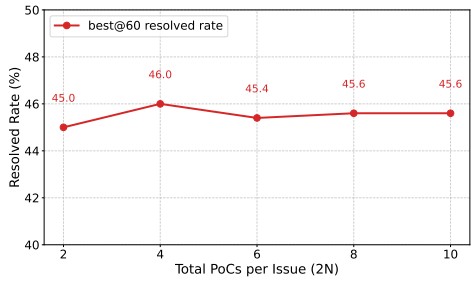

(a) The best@20 resolved rate for different training data/model sizes for validation.

(b) The best@60 resolved rate for # of PoCs.

Figure 8: More ablation studies on the validation component.

**Data size.** We randomly sample a subset of 500 and 1K from our 2K training set and train in total of three models for our *Val-no-assert* models training. We report the best@20 final resolved rate performance of these models in Figure 8a. The result shows that the performance keeps increasing from 500 to 1K and from 1K to 2K. As such, we select 2K as our final training data size.

**Model size.** We change our base model to Qwen-2.5-Coder-7B and Qwen-2.5-Coder-32B (same model family with different sizes) and retrain our *Val-no-assert* model with the same training data. The localization results in Figure 8a show that a larger model indeed improves the performance. However, considering that the improvement of the 32B model over the 14B model is not significant, we still choose the smaller one.

**Testing-phase scaling.** We reuse the same 60 candidate patches from the generation and vary only the number of PoCs supplied to the dynamic test. Specifically, we let both *Val-no-assert* and *Val-assert* generate N=1, 2, 3, 4, 5 PoCs each, yielding 2N=2, 4, 6, 8, 10 PoCs per issue. For every 2N, we measure the *best@60(The final resolved rate of the patches that chosen from 60 cases)* resolved rate after validation chooses a single patch from the 60 candidates. As shown in Figure 8b, increasing from N=1(2 PoCs total) to N=2 (4 PoCs) per model brings a clear boost, but adding more PoCs provides no further gains. The plateau occurs because the same validation model tends to emit highly similar PoCs; once two distinct checks are present, additional ones rarely catch new failures and instead lengthen the test phase without improving accuracy.

# D  Majority Voting vs. ORMs

**Design.** After dynamic testing, we often obtain several patch candidates that all pass the same subset of PoCs and functionality tests. Our default system selects the final patch via majority voting. To assess whether more reranking methods help, we keep every other component fixed—the 60 patch candidates, the four PoCs used for dynamic validation, and all hyper-parameters from 4.1—and only swap the tie-breaking strategy: *Majority Vote* (what we use). *ORM Score*, we feed each surviving candidate into the ORM model from SWE-Gym [36] and pick the highest-scoring patch. *Claude Vote*, we prompt Claude-3.7-Sonnet with the issue description, the localization results, and the patch candidates, and let it rank the patches; the top suggestion is selected. This setup isolates the impact of the tie-breaking itself, because every method starts from exactly the same candidate pool and the same dynamic-testing evidence.

**Results.** With 60 patching candidates, majority voting achieved the best accuracy (46.00% best@1), ahead of the ORM score (36.40%) and the Claude vote (45.00%). Because majority voting is cost-free(neither needs commercial credits nor computation resources), it remains the simplest and most effective option when many patches are still in play. Besides, its performance is better than even the SOTA commercial LLMs when dealing with a large pool, demonstrating its effectiveness.

Table 6: Effect of different tie-breaking strategies after dynamic testing.

| Final patch selection | best@60 (%) |
|---|---|
| Majority Vote | 46.00 |
| ORM Score | 36.40 |
| Claude Vote | 45.00 |

# E   Prompts for Each Component

## E.1   Localization Prompt

### E.1.1   File Localization Prompt

**User Prompt**:
Please look through the following GitHub problem description and Repository structure, and provide a list of files that one would need to edit to fix the problem.

### GitHub Problem Description ###
{problem_statement}
###

### Repository Structure ###
{structure}
###

After analyzing the problem, provide the full path and return at most {file_number} files. The returned files should be separated by new lines, ordered by most to least important, and wrapped with ```
For example:
```

file1.py
file2.py
```

### E.1.2   Line Localization Prompt

**User Prompt**:
Please review the following GitHub problem description and relevant files, and provide a set of locations that need to be edited to fix the issue.
The locations should include exact line numbers that require modification.
Pay attention! You should identify the method responsible for the core functionality of the issue. Focus on areas that define or enforce foundational behavior rather than case-specific issues.

### GitHub Problem Description ###
{problem_statement}

###
{file_contents}

###

{last_search_results}

After analyzing the problem, please provide the class name, function or method name, or the exact line numbers that need to be edited.

If you want to provide the line number, please give me a number in the middle every time.
If you need to edit multiple classes or functions, please provide all the function names or the line numbers in the class.
You should always include a class or function; do not provide just the line numbers without the class or function name.
If you want to include a class rather than a function, you should always provide the line numbers for the class.
Here is the format you need to strictly follow, don't return any code content or other suggestions, don't forget the "```":
### Examples:
```

full_path1/file1.py
class: MyClass1
line: 51

full_path2/file2.py
function: MyClass2.my_method
line: 12

full_path3/file3.py
function: my_function
line: 24
line: 156
```

## E.2 Generation Prompt

### E.2.1 Patch generation Prompt

**User Prompt**:
We are currently solving the following issue within our repository.

You are a maintainer of the project. Analyze the bug thoroughly and infer the underlying real problem, using your inherent knowledge of the project. Focus on resolving the root logic issue rather than suppressing symptoms.

Note that if the issue description mentions file names or arguments for reproduction, the fix must be generalized and not restricted to specific arguments. If the issue description includes a recommended fix, adapt it to align with the codebase's style and standards. Ensure your fix maintains structural integrity, considering interactions across code sections, nested structures, function calls, and data dependencies. Prefer solutions resilient to future structural changes or extensions.

The following is the issue description:

— BEGIN ISSUE —
{problem_statement}
— END ISSUE —

Below are the code segments from multiple files relevant to this issue. Each file is clearly marked. Decide carefully and only modify necessary segments. Preserve original indentation and formatting standards strictly.

— BEGIN FILES —
{content}

— END FILES —

Now, carefully analyze the files above. Determine which specific file segments require modifications and provide your edits using the following structured format for easy parsing:

<<< MODIFIED FILE: path/to/filename >>>
```python
<<<<<<< SEARCH
from flask import Flask
=======
import math
from flask import Flask
>>>>>>> REPLACE
<<< END MODIFIED FILE >>>
...
```

Please note that the *SEARCH/REPLACE* edit REQUIRES PROPER INDENTATION. If you would like to add the line ' print(x)', you must fully write that out, with all those spaces before the code!
Wrap the *SEARCH/REPLACE* edit in blocks ```python...```.

### E.2.2 Patch Critique Prompt

**User Prompt**:
Please continue working on this code patching work. You need to review the patch thoroughly to determine if it successfully fixes the issue without introducing any new bugs, and while handling all possible edge cases.
If you determine that the patch is correct and complete, tell me you confirm that the patch succeeded.
If you think the patch is incomplete, give me the reason and potential fixing suggestions.

You need to think:
1. What edge cases can break the patch? Consider complex cases such as nested structures and recursive patterns. For example, if the patch fixes an issue with an empty string, consider whether None, an empty list, or partially empty data structures might also trigger the bug.
2. Why the patch is incomplete or correct, whether the interaction between the patched part and other parts of the codebase can be handled properly
3. whether the patch only fixes the issue for the specific case mentioned in the issue description or for all similar cases
4. whether the patch follows the codebase's style and standards, using the proper variable types, error or warning types, and adhering to the established format

If the patch is perfect, tell me why. If the patch is unfinished or wrong, give me the reason and patch suggestions.
At the end, you should give me the critical result from you, if yes, give me "Conclusion: Right", otherwise give me "Conclusion: Wrong".

### E.3 Validation Prompt

### E.3.1 PoC Generation Prompt

**User Prompt**:
When generating a PoC script, follow these steps **in order**:

[Optional] Always include assertions in the PoC to make the failure obvious when the script is executed.
[Optional] The assertion should fail if the bug is present, and pass if the bug is not present.

**Try to extract an existing PoC from the issue description**
* Scan the **GitHub issue description** for Python code blocks or inline snippets that appear to reproduce the bug.
* If such a snippet exists, **use it verbatim as the base PoC** and only make the minimal edits needed to run:
- Remove interactive prompts (`>>>`, `$`, `In [ ]:`) and any captured output lines.
- Add any missing `import` statements.
- Convert Python2 syntax to Python3, if present.
- Merge multiple fragments into a single runnable file in their original order.

**If no valid PoC can be extracted, write one yourself**
* Use the *specific* classes, functions, or code paths named in the issue to trigger the bug.
* Keep the script minimal—just enough to demonstrate the failure (e.g., an `assert`, an expected exception, or a visibly incorrect result).

**General rules for both cases**
* The PoC **must be a single, self-contained Python3 file**.
* If the issue description includes other languages or shell commands, recreate their behavior in Python (e.g., with `subprocess` or file operations).
* If the snippet refers to external files, create them program matically inside the script.
* Always include `print()` or `assert` statements so the failure is obvious when the script is executed.

**Output format**
Return **exactly** Python code wrapped in triple backticks, with no other text.

```python
{{poc_code here}}
```
### Context Provided to You
{last_time_poc_code}

{execution_output}

### GitHub Issue Description
— Begin Issue Description —
{problem_statement}
— End Issue Description —

### E.3.2 PoC Critique Prompt

**User Prompt**:
You are a code reviewer for a GitHub project. Your task is to evaluate whether a patch

successfully resolves an issue described in a GitHub issue.

You will be given the issue description, the PoC (Proof of Concept) code that demonstrates the issue, and the PoC execution output before and after the patch.

Your goal is to determine if the patch resolves the issue. Please respond with "Yes" or "No" and provide a brief explanation of your reasoning.
You should not assume that there is other output that is not shown in the Poc Output. The Poc Output is the only output you should consider. You should not assume that a plot is generated by the Poc.

- "Yes" means the patch successfully fixed the issue.
- "No" means the patch did not successfully fix the issue, either because the issue still exists or because the patch introduced new issues.

### Raw Issue Description ###
{issue_description}

### Poc code ###
Here is the PoC code that demonstrates the issue:
{poc_code}

### Poc Output before the patch ###
{old_execution_output}

### Poc Output after the patch ###
{new_execution_output}

**Response Format:**

Example 1:
<judgement> Yes </judgement>
<explanation> The patch successfully fixed the issue since ... </explanation>
Example 2:
<judgement> No </judgement>
<explanation> The patch did not successfully fix the issue since ...
</explanation>

