# OpenReview forum: "Co-PatcheR: Collaborative Software Patching with Component-specific Small Reasoning Models"
_NeurIPS.cc/2025/Conference — NeurIPS 2025 poster_

### Official Review · Reviewer_awx7 · 2025-06-16

**Clarity:** 2
**Significance:** 3
**Originality:** 4
**Rating:** 5
**Confidence:** 4

**Summary:**

The paper investigates bug fixes.
While it is common to use a single model for the task, the presented method uses dedicated smaller models for sub tasks, and reaches a higher resolved rate with fewer overall parameters.

**Questions:**

“The key challenge is to design the patch-correctness validation.”
This is indeed a key challenge and I did not understand how you coped with it.
Note that in many fixes new tests are added for the fixed bug. These new tests fail on the original version and pass on the fixed one. Maybe you can identify such tests in your dataset and use them for validation.
Of course that a correct solution should pass the old tests too.


Part of the value of your model is being more efficient, represented by the number of parameters.
Presenting other resources (e.g., time) used for training and inference and their savings will make the benefit clearer.

If you will use the actual localization in the analysis of generation
You will factor the localization step out
You’ll be sure that there is a valid path in this localization
You might want to also use None LLM-based localization methods in the ablation.

Also note that localization has possible problems
In many cases there are some correct localizations (e.g., adding the same function in  one of possible locations).
Tangling commits, taking care of few tasks like a bug fix and refactoring, might include irrelevant modified code

**Ethical Concerns:**

["NO or VERY MINOR ethics concerns only"]

**Final Justification:**

I recommended to accpet in the first round and paper improved since.

**Limitations:**

Training details are given in references and the not provided appendices and code. Please complete them and describe them better in the paper itself.

Resolved definition should be given.

**Paper Formatting Concerns:**

“Where specialized engineers collaborate by dividing responsibilities according to their expertise” - usually engineers do not work like that

patching errors -> bug fixing

“As shown in Figure 4, in general, the performance of our models first increases as the training data grows and then remains at a similar level. Guided by this trend, we
select only 2,000 training issues, which is significantly fewer than existing RL-based methods, e.g., 11M in SWE-RL [50]” - Figure 4 does not refer to the number of samples. Also, it is rather strange that such a small data set is enough.

“For Val-assert model, we only train it to generate the PoCs, as the PoC correctness can be decided by assertions.”  - clarify where do the assertions come from

Appendices are not included in the file.

Could not find code and data

**Quality:**

3

**Strengths And Weaknesses:**

The paper suggests an Interesting direction.
The positioning is not clear, yet if this direction is new, it can be generalized to many areas.

The paper does ablation analysis, which contributes to the understanding of each component.
Using the actual location can help for a cleaner evaluation of generation.

It is not clear how “resolved” is decided? Comparison to actual fix? Passing the dataset tests? The generated tests? Please specify.


In the same point, the roles of the Val-no-assert model and Val-assert model are not clarified.
Are they just critiques, helping to improve the generation or deciding the  “resolved” classification?
If so, their failure might raise the resolved rate .
Similarly, it is not clear enough how the expected result and loss function are used in training, if such training is done.

All specialized models perform dramatically worse than the general purpose models, making the direction interesting but not very useful.

---

> ### Author Rebuttal · Authors · 2025-07-30
>
> Thank the reviewer for the constructive and positive comments. Please see below for our responses.
>
> ## 1. Definition of “Resolved”
>
> Sorry for the confusion. We follow the standard SWE-bench program setup for issue resolving: Given only the issue description of a target issue and the entire codebase as the input, the patching tools should output a patch diff as an answer automatically.
>
> Generally, the design of patching tools will contain three main modules to handle such tasks: (1) localize the problematic code regions, (2) generate a patch to fix the issue, and (3) validate the patch through our own validation module before submitting the final repair (given that test cases for the issues are not available).
>
> "resolved" means that, after applying our generated patches to the original codebase using Unix's patch program, SWE-bench will execute the existing unit and system tests (including the new golden test for the current issue) associated with that repository. A patch is considered "resolved" if it applies successfully and all existing tests pass. These test cases are collected by the SWE-bench and are not available to any patching tools during the issue resolving process. We acknowledge that even though the patched version passes all the tests, it is still possible that the patch introduces new bugs. We believe it is an interesting future direction to conduct more comprehensive evaluations of patch correctness and security beyond standard unit tests (e.g., using fuzzing tools).
>
> ## 2. Roles of the Validation Model
> Sorry for the confusion. The main purpose of our PoC testing is to synthesize test cases for validating whether a candidate patch truly fixes the bug described in the issue. Both assertion tests and non-assertion tests serve this purpose, but in different styles.
>
> The assertion tests **independently determine whether the bug is triggered**, whereas non-assertion tests **only record the output of the PoC and require an LLM to judge the output after execution**.
> - **Assertion tests** follow the common unit test style, which would provide an explicit oracle to the target issue—for example, assert sort([2,1]) == [1,2]. When such a test fails, we know that the bug is highly likely to still exist. However, the downside is that writing a correct oracle requires the model to infer an exact post-condition for the issue, which can be difficult..
> - **Non-assertion tests** relax this requirement by performing a general PoC script. These are similar to differential testing, where we compare execution outputs before and after applying the patch to determine fix effectiveness. They are generated to a PoC according to the issue description, being expected to reproduce the same observable symptoms. In the downstream workflow, we would ask LLM to make a judgment according to the original issue description and the output of non-assertion tests. And another benefit from non-assertion tests rather than assertion tests is that non-assertion tests can capture the new bug introduced by the candidate patch by observing and analyzing actual output behavior.
>
> So we keep both kinds of tests for two reasons:
> - **Coverage** –  Some bugs have well-defined expected values or exceptions that can be explicitly tested with assertions, while others manifest as subtle behavioral changes or output variations that are better captured through execution observation rather than rigid assertion checks. Using both styles ensures we can validate a wider range of issues.
> - **Regression** – A patch might solve the original failure but introduce a new one. Because non-assertion tests observe the output of the program, they can detect these new bugs even though the old assertion still passes. The whole patching system can utilize this information for further patch refinement.
>
>
> ## 3.  Localization module explanation
> - **Factoring out the localization step**: We agree with the reviewer that using actual localization from patch commits could help isolate generation performance. However, it's important to note that even commit-based localization cannot guarantee completeness, as there is no true "golden localization" in this task setting. For effective code repair, models need sufficient contextual information beyond just the exact modified lines - including surrounding code, dependencies, and related functions. Additionally, the same bug can often be fixed through multiple valid approaches at different locations, making even commit-based localization an approximation rather than a definitive ground truth.
> - **Controlled evaluation**: In our experiments, we do use the same localization results across different generation models for comparison(in section §4.2.2, we use the same localization results from GPT-4o as the input of generation evaluation ), which effectively factors out localization variance and allows us to isolate the generation capabilities of different approaches.
> - **Tangling commits concern**: We agree with the reviewer that commits containing irrelevant modifications are due to mixed bug-fixing and refactoring activities. However, our work addresses this issue during training data collection. We address this by selecting training datasets that have already been filtered for issues primarily focused on bug fixes with corresponding unit tests, thereby minimizing the impact of irrelevant code modifications from tangling commits. This curation process ensures that our training data better reflects targeted bug-fixing scenarios rather than complex, multi-purpose commits.
> - **Non-LLM localization methods**: We acknowledge the suggestion to include non-LLM localization methods in our ablation study. However, our localization input only contains issue descriptions and repository code, where the issue descriptions often lack perfect proof-of-concept examples or precise error locations. This requires semantic understanding to map natural language problem descriptions to relevant code sections, making it difficult to effectively compare with traditional non-LLM localization methods that typically rely on explicit error traces, stack traces, or other structured debugging information.
>
>
> ## 4. Inference efficiency metrics
> In order to show the inference metric of our model and system, we have added an extra experiment to record the inference efficiency metrics, as shown in the following table. We log four kinds of metrics for an average of every issue: end-to-end wall-clock time for our testing setup in §4.1, tokens generated across all model calls (completion), latency for average one-shot model inference, and peak GPU memory. Experiments run on a pair of NVIDIA L40S (48 GB) cards with vllm deployment framework. The throughput can be further improved through parallel processing of multiple instances.
>
> |     Metric (per issue)     | Value (mean) |
> |:--------------------------:|:------------:|
> | End-to-end wall-clock time |    334.17 s    |
> |     Model-only latency     |    44.01 s    |
> |   Tokens generated   |    5.83 k    |
> | Peak GPU memory (per GPU)  |     44.64 GB    |
>
>
> ## 5. Explanation of the efficiency of a small training dataset
> Sorry for the issue with the figure reference, will correct it. Regarding the relatively small dataset size, there are several key reasons why our approach requires significantly fewer training samples compared to methods like SWE-RL:
> - **Knowledge Distillation**: Our model training employs supervised fine-tuning (SFT) through knowledge distillation from a teacher model, which enables efficient learning of the teacher's response structure and patterns. In contrast, RL-based methods like SWE-RL require extensive exploration and trial-and-error processes across many samples to optimize their policies, making them inherently data-hungry.
> - **Reasoning-Based Training Data**: Our training approach focuses on reasoning data that teaches the model to analyze problems systematically before generating solutions. This reasoning-first methodology allows the model to quickly internalize the teacher's logical reasoning patterns, leading to more efficient knowledge transfer compared to methods that learn from raw input-output pairs without explicit reasoning steps.
> - **Targeted Learning Objective**: Unlike other methods that need to explore vast action spaces, our approach has a more focused learning objective - replicating the teacher model's structured reasoning process for patch generation. This specificity reduces the data requirements significantly.
>
>
> ## 6. Training method
> All three models in our paper were trained using knowledge distillation from teacher models through supervised fine-tuning (SFT). The SFT training optimizes the standard cross-entropy loss between the model's predicted token probabilities and the ground truth tokens from the teacher-generated responses.
>
>
> ## 7. Typos and missing appendix and code
> Sorry for the typos, will correct them (e.g., “figure 4” should be “section 4”). We submitted the appendix in the supplementary material. We will also make our code, data, and models public.

---

> > ### Comment · Reviewer_awx7 · 2025-08-04
> >
> > 1.
> > Your definition of resolved is common and solid. I suggest clarifying that in the paper.
> > 2.
> > This can be a contribution of its own.
> > Adding a direct estimation of the contribution of the non-assertion tests might help.
> > 4.
> > To support your claim, you should add the metrics for the benchmark solution too (not critical, but can contribute to your solution value).
> >
> > I'm OK with the other comments.

---

> > > ### Author Response · Authors · 2025-08-05
> > >
> > > Thanks for reviewing our rebuttal. We are happy that the rebuttal can address the reviewer's concerns.
> > >
> > > 1. Yes, we will clarify this in the paper.
> > >
> > > 2. Thank you for the suggestion. The following table directly shows the effectiveness of no assertion only tests (Co-PatcheR-NoAssertOnly vs. Co-PatcheR-NoPoC). We will emphasize this point in the paper.
> > > | Validation Method | Resolved Rate (%) |
> > > |:--------------------------:|:------------:|
> > > | Co-PatcheR-NoPoC | 37.0 |
> > > | Co-PatcheR-NoAssertOnly | 42.6 |
> > >
> > >
> > > 4. Thank you for the suggestion. Most related works have not open-sourced their models, making direct efficiency comparisons challenging. Therefore, we compare our approach with SWE-Fixer(72B), one of the few available baselines. Experiments run on two NVIDIA L40S (48 GB) cards with the VLLM deployment framework. The comparison results are shown in the following table:
> > >
> > > | Metric (per issue) | Co-PatcheR | SWE-Fixer |
> > > |:--------------------------:|:----------:|:----------:|
> > > | End-to-end wall-clock time | 334.17 s | 123.25 s |
> > > | Model-only latency | 44.01 s | 103.08 s |
> > > | Tokens generated | 5.83 k | 2.51 k |
> > > | Peak GPU memory (per GPU) | 44.64 GB | 45.23 GB |
> > > | Resolved Rate (%) | 46.0 | 32.8 |
> > >
> > > SWE-Fixer achieves faster end-to-end execution primarily because its testing approach relies mainly on prompting the model without dynamic validation, while our framework includes comprehensive PoC testing and validation workflows. However, this trade-off results in significantly lower success rates (32.8% vs 46.0%). SWE-Fixer shows higher model-only latency due to their larger model size, while our reasoning-based approach generates more tokens due to the complexity of our framework and explicit reasoning steps. Despite the longer execution time, our method demonstrates higher efficiency in terms of successful patch generation per unit of computational cost.

---

> > > > ### Comment · Reviewer_awx7 · 2025-08-05
> > > >
> > > > 3.
> > > > I agree that for many use cases your higher resolved rate compensates for the slower time.
> > > > Just acknowledge that in the paper since these might be at least as important metrics as the number of parameters.

---

> ### Author Response · Authors · 2025-08-05
>
> Thank you for the feedback. We would also like to quickly mention that most methods require dynamic testing, which should have a similar computational time as ours. We will add this acknowledgment to the paper. Thanks!

---

### Official Review · Reviewer_ZdqA · 2025-07-02

**Clarity:** 2
**Significance:** 2
**Originality:** 3
**Rating:** 4
**Confidence:** 4

**Summary:**

This paper presents Co-PatcheR -- an approach to train smaller task-specific LLMs to target software engineering tasks. The authors focus on designing smaller training data for specific tasks like localization, patch generation, and test generation to train smaller local models. The core technique involves using the smaller models to complete individual tasks and finally solve the end-to-end task.

**Questions:**

1. Did the authors try using two separate models for patch generation and localization and evaluating the performance?
2. What is and what are the benefits of using with and without assertion tests?
3. How are multiple tests generated in order to evaluate the patches for one issue and how does the patch selection process work?

**Ethical Concerns:**

["NO or VERY MINOR ethics concerns only"]

**Final Justification:**

After reading the author replies and also additional reviews by the other reviewers, while I still hold some concerns (i.e., why repair and localization do not really benefit from separately training two models) But I believe that overall the paper fits the standard for neurips. As such I recommend acceptance and have increased my score

**Limitations:**

Yes

**Paper Formatting Concerns:**

No formatting issue

**Quality:**

2

**Strengths And Weaknesses:**

## strengths
- Deals with a very important problem of automatically solving software development problems through the use of specialized software agents
- The idea of using smaller distilled models instead of only using foundational sota models is a good direction to expore

## weaknesses

Overall, while I like the direction of the approach proposed in this work, I believe the technique and evaluation can be carried out more rigorously and would benefit from some more revision

Lack of separation between patch generation and localization model:
- One major surprising part of the work is the lack of separately trained localization and patch generation models (i.e., the authors train one joint model for both localization and patch generation)
- Localization and patch generation seems like to me two separately tasks (at least separate enough to warrant the construction different training datasets)
- The authors did not seem to evaluate this potential in the paper and only allude to the results in the introduction (I also don't see any discussion or results in the appendix)
- Furthermore, the authors also did not evaluate the evaluate the simple baseline of training a single model to accomplish all the tasks in order to demonstrate the importance of using separate models to solve this complex task

Val-assert/no-assert models:
- Its very unclear what with and without assertions means and how it may help to solve the particular issues, the author provides very limited description and justification for this seemly important component of the approach (as it involved 2 separate models)
- Are the tests with assertion meant to simply verify that the issue is present, and the non-assertion tests used to indicate that the issue has been resolved?
- Even after looking at the used prompt in appendix D this was not clear
- Furthermore, this statement "First, existing patching agents have two ways of generating PoC tests: with or without assertions." is provided with no citations or proofs. From my understanding I don't think any of the current agent-based approaches specifically distinguished between the described PoC tests.
- Additionally "... and 4 PoCs, using them to get one final patch" how are multiple tests used to get one final patch, and why would there be 4 tests in total for each issue?

One small suggestion is regarding Table-1:
- Since the authors compare against some of the prior approaches, it is important to acknowledge the different training methods
- So for example, the training dataset used in this is constructed by distill some larger models whereas other works like Agentless-mini from my understanding does not require the use of any larger llm and mainly rely on RL-based models

---

> ### Author Rebuttal · Authors · 2025-07-30
>
> Thank the reviewer for the constructive comments. Please see below for our responses.
>
> ## 1. Separation model between localization and patch generation
> There are two reasons why we combine the localization and generation in one model:
>
> - **Share Core Abilities**: Both subtasks require fundamental capabilities in understanding the issue description to identify what needs to be fixed, and reasoning over the current codebase to determine the appropriate intervention points and methods.
> - **Task Interdependence**: A better understanding of how to fix a bug helps clarify what context needs to be retrieved for localization, while precise localization provides essential context for accurate patch generation. Localization applies these abilities to locate faulty lines; generation applies the same context to edit them. Because the tasks are sequential, training them together allows the model to reuse intermediate reasoning and to learn a smoother “find-then-fix” workflow without growing the parameter count.
>
> We further experimented with two separate models—one fine-tuned for localization, the other for generation. Testing the results of pass@20(whether a correct patch is in the 20 generated candidates) on the SWE-bench Verified split as we did in ablation studies, to isolate the framework factor. The scores were 55.40% for the split setup and 56.20% for the joint model. The joint model performed at least as well, sometimes better.
>
> ## 2. Single model for all the tasks
> We also tried to explore the solution of training a single model for all the tasks in the patching process. A single 14 B model was fine-tuned jointly on localization, patch generation, and validation, and we re-evaluated it under the same best@20 on SWE-bench-Verified with the same hyperparameters in ablation experiments(5 root causes from localization, 20 patches from generation, and 4 PoCs from validation). We can see that the
>
> | Model                  | File-loc  | Line-loc  | Pass@20  | best@20  |
> |------------------------|-----------|-----------|---------------|---------------|
> | All-in-One 14 B    | 85.20%    | 51.60%    | 55.20%       |    41.40%    |
> | Co-PatcheR (3×14 B)    | 85.00%    | 51.80%    | 56.20%    | 43.20%    |
>
> Although the All-in-One model matches Co-PatcheR on the two upstream metrics— localization(file-level and line-level), Pass@20 resolved rate for patch generation and its best@20 resolved rate are worse than Co-PatcheR.
>
> This gap appears in the validation stage. For the Localization & Generation task, the model shares common capabilities as demonstrated in part ##1, and benefits from consistent, stable behavior. However, the validation module, especially the test generation part, requires the ability to produce diverse and comprehensive tests to thoroughly explore edge cases and corner scenarios. When both objectives are packed into a single set of model weights, it creates a fundamental tension: maintaining consistency for localization/generation pushes the model toward conservative behavior, which conflicts with the need for creative and diverse test generation in validation. Consequently, the model produces overly narrow or insufficient test suites that fail to adequately validate patches. This means that potentially correct patches—which would pass validation if tested more comprehensively—are incorrectly rejected due to incomplete testing coverage, ultimately reducing the overall bug resolution rate.
>
>
> ## 3.  Assertions and non-assertions testing
> Sorry for the confusion. The main purpose of our PoC testing is to synthesize test cases for validating whether a candidate patch truly fixes the bug described in the issue. Both assertion tests and non-assertion tests serve this purpose, but in different styles.
>
> The assertion tests **independently determine whether the bug is triggered**, whereas non-assertion tests **only record the output of the PoC and require an LLM to judge the output after execution**.
> - **Assertion tests** follow the common unit test style, which would provide an explicit oracle to the target issue—for example, assert sort([2,1]) == [1,2]. When such a test fails, we know that the bug is highly likely to still exist. However, the downside is that writing a correct oracle requires the model to infer an exact post-condition for the issue, which can be difficult..
> - **Non-assertion tests** relax this requirement by performing a general PoC script. These are similar to differential testing, where we compare execution outputs before and after applying the patch to determine fix effectiveness. They are generated to a PoC according to the issue description, being expected to reproduce the same observable symptoms. In the downstream workflow, we would ask LLM to make a judgment according to the original issue description and the output of non-assertion tests. And another benefit from non-assertion tests rather than assertion tests is that non-assertion tests can capture the new bug introduced by the candidate patch by observing and analyzing actual output behavior.
>
> So we keep both kinds of tests for two reasons:
> - **Coverage** –  Some bugs have well-defined expected values or exceptions that can be explicitly tested with assertions, while others manifest as subtle behavioral changes or output variations that are better captured through execution observation rather than rigid assertion checks. Using both styles ensures we can validate a wider range of issues.
> - **Regression** – A patch might solve the original failure but introduce a new one. Because non-assertion tests observe the output of the program, they can detect these new bugs even though the old assertion still passes. The whole patching system can utilize this information for further patch refinement.
>
> ## 4. Multiple test usage in the validation module
> Sorry for the confusion. We rank patches based on dynamic testing results (number of passed PoC and functionality tests) to identify the highest-scoring candidates. When multiple patches achieve the same highest score, we apply majority voting to select the final patch. Will clarify this in the paper.
>
> As for the reason why we need 4 PoC tests, without the golden test case about the issue, we need to synthesize related test cases to validate whether a candidate patch truly fixes the bug described in the issue. Given that we can not make sure one-shot test generation can exactly describe the issue, we would like to increase the testing cases to have better coverage and robustness as mentioned above. And we also have an ablation study in §Appendix B.3 to show why we use 4 PoC tests in our system.
>
> ## 5. Citations of PoC test generation classification
> We acknowledge that existing methods do not explicitly categorize and emphasize the distinction between assertion and non-assertion tests. However, through our analysis of actual project implementations, we observed that Agentless[1] generates assertion-based reproduction tests for validation, while PatchPilot[2] produces non-assertion tests and employs LLMs for test result evaluation. We will make sure to clarify this in the paper.
>
> ## 6. Suggestions for Table 1
>
> Thanks for the valuable suggestion. We will update Table 1 with more information regarding the differences between our method and the baselines (e.g., distillation vs. RL).
>
>
> [1] Agentless: Demystifying LLM-based Software Engineering Agents
> [2] PatchPilot: A Cost-Efficient Software Engineering Agent with Early Attempts on Formal Verification

---

> > ### Author Response · Authors · 2025-08-06
> >
> > We sincerely hope our responses have adequately addressed your concerns and appreciate your valuable feedback. We would appreciate any further suggestions or feedback. Thanks!

---

> > > ### Comment · Reviewer_ZdqA · 2025-08-06
> > >
> > > Firstly thank the authors for the detailed replies to my questions and concerns
> > >
> > > >Separation model between localization and patch generation
> > >
> > > I agree with the authors that the two tasks share similar utilities and "inputs"/"reasonings". However, it is still definitely two different tasks. In fact there is a large amount of literature which targets these two different problems seperately. I understand the score obtained by the authors are lower when the models are not trained jointly. However, I feel the explanation and perhaps more evaluation might be needed for theses things to better understand it
> > >
> > > >Single model for all the tasks
> > >
> > > It seems the scores are especially close with just using one model (e.g., 3x less size) compared to using three models. Still I appreciated this more detailed comparison
> > >
> > > >Assertions and non-assertions testing
> > >
> > > Based on the author's own descriptions, both tests are basically the same since we can simply observe the output of both tests to determine if the bug is triggered. The author claims that "Because non-assertion tests observe the output of the program, they can detect these new bugs even though the old assertion still passes." but this is also true for assertion tests, you can still observe the output of assertion test (apart from just the assertion) to determine if there are some regression failure
> > >
> > > >Overall
> > >
> > > Again thanks for the detailed replies, although I still hold some concerns regarding this work, I will not fight for the rejection. As such i have increased my score.

---

> > > > ### Author Response · Authors · 2025-08-07
> > > >
> > > > We sincerely thank the reviewer for the constructive and supportive feedback. We will include rebuttal experiments, our discussion in our paper. Below, we would like to provide some quick follow-ups.
> > > >
> > > > Regarding point 1, thank the reviewer for acknowledging the shared capabilities between the two tasks. We totally agree with the reviewer that these are different tasks and substantial future work is needed to fully investigate whether these two models can be merged or not. Some concrete directions: can try it on more languages like C/C++ with complex memory operations, or security bugs where crash location can be different from the root causes.
> > > >
> > > > Regarding point 3, we agree with the reviewer that from the perspective of the regression test, these two points serve similar capabilities. Empirically, we observe that these approaches generate tests with different patterns and styles. Will clarify in the paper.
> > > >
> > > > Thanks!

---

### Official Review · Reviewer_GmxY · 2025-07-03

**Clarity:** 2
**Significance:** 3
**Originality:** 3
**Rating:** 4
**Confidence:** 3

**Summary:**

The authors propose decomposing the task of software repair into explicit steps: localisation, patch/repair, and validation. Furthermore, they argue that the expected capabilities and inputs at each stage can vary, and thus propose using different models tailored to each step. They identify two sets of requirements: for localisation and generation, the model requires access to the issue description and entire code base (especially for localisation); for validation, the model requires a deeper understanding of the issue description; however, it can forgo access to the full code base, instead focusing on the method under test. Further, they argue that in the interest of a fairer validation, the validation model should not have access to patch candidates. To train their models, the authors select problems across repositories that do not overlap with their test set and that over a range of problem difficulty levels. They then design and break down the problems at each phase. For localisation, they break down the task into file localisation and line localisation. For generation, they integrate patch refinement by combining the generation step with a critique step that provides feedback. For validation, they propose to generate unit tests with and without assert statements, arguing that this facilitates testing a wider range of program behaviours. The proposed final tool, Co-PatcheR, manages to achieve competitive results (compared to some commercial models) and improves over specialised open-source models at a greatly reduced sample cost, achieving a 46% resolved rate on SWE-bench-verified using 6K training examples.

**Questions:**

Q1 (repeat from W2)  It is unclear what the authors mean by assert and assert-less tests. Is checking the expected outcome considered an assert? If yes, what _exactly_ are the assert-less tests verifying, program crashes? The lack of clarity around writing these details confused me regarding the justification of the two validation models, as it is unclear what kind of output they produce. Further, it is unclear to this reviewer how writing unit tests that have asserts fundamentally covers different program paths than those that do not; is the latter trying to approximate fuzzing?

Q2 I remain unconvinced by the merging of Localisation and Generation. The Generation models would require different types of contexts compared to localisation: related/co-changed files, called files, a table of available symbols (APIs), etc. These feel materially different from the full code-base view of Localisation unless Localisation is reused to find the relevant files/symbols. Was a set-up where the two stages use different models considered? If not, is it due to a higher priority during model design being given to reducing model size/parameter count?

- Nit: Page 4, Line 138, you refer to test cases as vulnerable? Is this from an older iteration of the paper where the focus was on security testing? Also, same location, you introduce vulnerable test cases as PoC (proof of concept?), this seems highly unusual and does not match the words abbreviated.
- Writing Nit: §3.2.1 "following recent work by S1 [31]" -> "following recent work in S1 [31]" or "following recent work by Muennighoff et al. [31]" as "S1" is the tool/method name.

**Ethical Concerns:**

["NO or VERY MINOR ethics concerns only"]

**Final Justification:**

The authors have addressed my concerns in the rebuttal phase, and I believe this paper is of NeurIPS quality. I maintain my positive score.

**Limitations:**

Limitations are reasonably addressed.

**Paper Formatting Concerns:**

N/A, paper seems to follow formatting guidelines.

**Quality:**

3

**Strengths And Weaknesses:**

+ S1 Strong results on SWE-bench-verified at a considerably reduced sample cost.
+ S2 Modulo the remark for W2 below, justified decomposition into sub-problems and sub-models.
+ S3 Strong results on sub-problems (localisation), not just the end-to-end task
- W1[writing] The paper makes claims about testing in general (page 4, lines 125-127), arguing that testing does not require access to the full code base. This claim is true for unit testing, which is the scenario in the paper (though never stated explicitly), but it is not true in all testing settings: acceptance, integration, interaction testing as three examples require access to multiple units and in an end-to-end integration test suit, knowledge of the full code-base would be required. The problem setting should be clarified to be that of unit testing.
- W2 It is unclear what the authors mean by assert and assert-less tests. Is checking the expected outcome considered an assert? If yes, what _exactly_ are the assert-less tests verifying, program crashes? The lack of clarity around writing these details confused me regarding the justification of the two validation models, as it is unclear what kind of output they produce. Further, it is unclear to this reviewer how writing unit tests that have asserts fundamentally covers different program paths than those that do not; is the latter trying to approximate fuzzing?

---

> ### Author Rebuttal · Authors · 2025-07-30
>
> We thank the reviewer for the positive and constructive comments. Please see below for our responses.
>
> ## 1. Clarification of unit testing
> Thank you for pointing this out. Yes, our work focuses exclusively on unit-test–level validation, where each generated test exercises a single function or script in isolation. We will make sure to clarify this in the next revision to avoid confusion with other test levels (integration, acceptance, etc.).
>
> ## 2. Difference between assertion tests and non-assertion tests
> Sorry for the confusion. The main purpose of our PoC testing is to synthesize test cases for validating whether a candidate patch truly fixes the bug described in the issue. Both assertion tests and non-assertion tests serve this purpose, but in different styles.
>
> The assertion tests **independently determine whether the bug is triggered**, whereas non-assertion tests **only record the output of the PoC and require an LLM to judge the output after execution**.
> - **Assertion tests** follow the common unit test style, which would provide an explicit oracle to the target issue—for example, assert sort([2,1]) == [1,2]. When such a test fails, we know that the bug is highly likely to still exist. However, the downside is that writing a correct oracle requires the model to infer an exact post-condition for the issue, which can be difficult..
> - **Non-assertion tests** relax this requirement by performing a general PoC script. These are similar to differential testing, where we compare execution outputs before and after applying the patch to determine fix effectiveness. They are generated to a PoC according to the issue description, being expected to reproduce the same observable symptoms. In the downstream workflow, we would ask LLM to make a judgment according to the original issue description and the output of non-assertion tests. And another benefit from non-assertion tests rather than assertion tests is that non-assertion tests can capture the new bug introduced by the candidate patch by observing and analyzing actual output behavior.
>
> So we keep both kinds of tests for two reasons:
> - **Coverage** –  Some bugs have well-defined expected values or exceptions that can be explicitly tested with assertions, while others manifest as subtle behavioral changes or output variations that are better captured through execution observation rather than rigid assertion checks. Using both styles ensures we can validate a wider range of issues.
> - **Regression** – A patch might solve the original failure but introduce a new one. Because non-assertion tests observe the output of the program, they can detect these new bugs even though the old assertion still passes. The whole patching system can utilize this information for further patch refinement.
>
> ## 3. The combination of localization and patch generation
> There are two reasons why we combine the localization and generation in one model:
>
> - **Share Core Abilities**: Both subtasks require fundamental capabilities in understanding the issue description to identify what needs to be fixed, and reasoning over the current codebase to determine the appropriate intervention points and methods.
> - **Task Interdependence**: A better understanding of how to fix a bug helps clarify what context needs to be retrieved for localization, while precise localization provides essential context for accurate patch generation. Localization applies these abilities to locate faulty lines; generation applies the same context to edit them. Because the tasks are sequential, training them together allows the model to reuse intermediate reasoning and to learn a smoother “find-then-fix” workflow without growing the parameter count.
>
> We further experimented with two separate models—one fine-tuned for localization, the other for generation. Testing the results of pass@20(whether a correct patch is in the 20 generated candidates) on the SWE-bench Verified split as we did in ablation studies, to isolate the framework factor. The scores were 55.40% for the split setup and 56.20% for the joint model. The joint model performed at least as well, sometimes better.
>
> ## 4. Inaccurate phrasing
> Thank you for spotting the typo on page 4, line 138. We will remove the “vulnerable” description because we are not only focusing on security problems, and we will also clarify PoC as ‘proof-of-concept’ to avoid ambiguity. And we will update the correct citation name in §3.2.1.

---

> > ### Comment · Reviewer_GmxY · 2025-08-04
> >
> > I thank the authors for the detailed response.
> >
> > I think 1, 3, and 4 clearly address my questions.
> >
> > Regarding 2., I think the authors and I work under different definitions of "oracle", `asserts' being an example thereof. I was considering a golden program A under regression testing to be ``effectively'' an `assert', hence the initial confusion. From the response, it is clear that the authors do not operate under such an assumption. If possible, in the appendix, I would like to see a better explanation of this differential testing setup and perhaps some example observables under watch (or if you use the full heap/stack, stating it as such).
> > Generally, I agree with the authors, as stated in their response, that both coverage and validation (of the new feature via assertion tests) are important to assess correctness, as well as regression testing to ensure the non-introduction of new bugs.

---

> > > ### Author Response · Authors · 2025-08-04
> > >
> > > Thanks for reviewing our rebuttal. We are happy that the rebuttal can address the reviewer's concerns.
> > >
> > > Sorry for the confusion regarding the second point - we will follow the reviewer's suggestion and add a more detailed explanation in the appendix to clearly differentiate between our assertion tests and non-assertion tests, along with concrete examples.

---

> > > > ### Comment · Reviewer_GmxY · 2025-08-04
> > > >
> > > > Reading some of the other responses so far, I think others also see a potential contribution in this as well, and I would agree. Unit tests, Regression testing, Functional/Non-Functional are aspects discussed in SE, but I do not see that often in ML, and the Regression aspect is important!

---

> > > > > ### Author Response · Authors · 2025-08-05
> > > > >
> > > > > Thank you for the positive feedback and for highlighting the importance of regression testing in the context of ML-based patch generation. We agree that bridging software engineering testing methodologies with ML approaches is a valuable contribution that deserves more attention in the field. We regard these as our future work and will mark them down in our paper later, thanks!

---

### Official Review · Reviewer_ggJ3 · 2025-07-04

**Clarity:** 3
**Significance:** 3
**Originality:** 3
**Rating:** 4
**Confidence:** 4

**Summary:**

This paper introduces Co-PatcheR, a collaborative patching framework composed of small, component-specific reasoning models for software patching. The authors propose decomposing the patching pipeline into three components: localization, patch generation, and validation. Co-PatcheR uses a single Loc-Gen model for both localization and generation, and two separate models for diverse issue-reproducing test case (PoC) generation and validation. The authors design tailored task formulations and training recipes for each component, including a lightweight reasoning data construction method and a model distillation strategy. Experimental results on the SWE-bench-Verified benchmark show that Co-PatcheR achieves a 46% resolved rate using only 3×14B models and 6K training samples, outperforming larger baselines while being substantially more parameter- and data-efficient.

**Questions:**

Q1. Could the authors report cross-framework results (e.g., Co-PatcheR + Agentless vs. SWE-RL-70B + Agentless, and Co-PatcheR + PatchPilot vs. SWE-RL-70B + PatchPilot) to better disentangle the effects of model architecture and agent framework?

Q2.Could the authors provide inference efficiency metrics (e.g., latency, total generated tokens, or wall-clock time) to assess practical deployment cost?

**Ethical Concerns:**

["NO or VERY MINOR ethics concerns only"]

**Final Justification:**

The authors’ response has adequately addressed my concerns. Taking into account the comments from other reviewers and the corresponding responses from the authors, I have decided to keep my rating at 4.

**Limitations:**

Yes

**Quality:**

3

**Strengths And Weaknesses:**

Strength
1. The idea of decomposing patching tasks into collaborative reasoning components, each trained with customized objectives and data, is practically valuable, especially in resource-constrained scenarios.
2. The method achieves a 46% resolved rate on SWE-bench-Verified, surpassing large end-to-end baselines while using fewer parameters and less data.
3.The authors provide extensive ablations that validate their architectural and training choices, such as the joint Loc-Gen model, multi-model PoC generation, critique-based generation, and data/model scaling considerations.

Weakness
1. Given that, when using the same model (Claude-3.5), the agent framework PatchPilot achieves better performance than the agentless setting, it would be valuable for the authors to conduct additional cross-framework evaluations. For example, comparing Co-PatcheR + Agentless with SWE-RL-70B + Agentless, or Co-PatcheR + PatchPilot with SWE-RL-70B + PatchPilot. Such comparisons would help disentangle the respective contributions of the model architecture and the agent framework, providing fairer and more convincing evidence to support the authors’ insight that multiple expert small models can outperform a large end-to-end model on SWE-bench.
2. The authors present an encouraging result that three small models, requiring fewer parameters and less training data, can outperform a large end-to-end model on SWE-bench. However, beyond training cost, inference cost is equally critical for practical applications. The current study primarily focuses on generation quality metrics (e.g., pass@1, pass@K) without reporting inference time or the number of generated tokens. In deployment scenarios, inference latency and token generation efficiency are important considerations — particularly for approaches like Co-PatcheR, which rely on generating large numbers of candidate patches to match the performance of larger models. It would strengthen the paper to include an evaluation or at least a discussion of inference efficiency alongside generation accuracy.

---

> ### Author Rebuttal · Authors · 2025-07-30
>
> We thank the reviewer for the positive and constructive comments. Please see below for our responses.
>
> ## 1. Cross-framework evaluation experiments
>
> Following the reviewer’s suggestion, we integrate the Co-PatcheR models into the agentless framework. We measure best@60 on the SWE-bench-Verified benchmark as a balance of performance and efficiency. All other hyperparameters are kept identical to SWE-RL to enable an apple-to-apple comparison.
>
> | Method | Resolved Rate |
> |:------:|:-------------:|
> | Co-PatcheR + PatchPilot | 46.00% |
> | Co-PatcheR + Agentless | 45.20% |
> | SWE-RL-70B + Agentless | 41.00% |
>
> Even with the changing of the testing framework, Co-PatcheR keeps a 4.20% advantage over the 70B SWE-RL model final results, showing the strengths of Co-PatcheR models.
>
>  We also observe a 0.80% drop from Co-PatcheR+PatchPilot (46.00%) to Co-PatcheR + Agentless. This is because the models are trained based on the agentic data generated using the PatchPilot framework, which are naturally more suitable for use in PatchPilot. Besides, the validation strategy in PatchPilot and Agentless is different: for example, we utilize two kinds of PoCs(assertion and non-assertion) in PatchPilot, which also causes some performance discrepancy.
>
> Also, we would like to clarify that SWE-RL has not released its weights, so we cannot re-run it in any agent framework. That’s why we can not do SWE-RL + PatchPilot testing, but only compare with the results they mentioned in their paper.
>
> ## 2. Inference efficiency metrics
>
> Following the reviewer’s suggestion, we comprehensively evaluated the efficiency of Co-PatheR, as shown in the following table. We log four kinds of metrics for an average of every issue: end-to-end wall-clock time for one instance under our testing setup in the paper(5 root causes from the localization model and 4PoCs from the validation model), tokens generated across all model calls (completion), latency for average one-shot model inference, and peak GPU memory. These metrics provide the efficiency baseline for Co-PatcheR on two NVIDIA L40S (48 GB) cards with a VLLM deployment framework. The throughput can be further improved through parallel processing of multiple instances.
>
> |     Metric (per issue)     | Value (mean) |
> |:--------------------------:|:------------:|
> | End-to-end wall-clock time |    334.17 s    |
> |     Model-only latency     |    44.01 s    |
> |   Tokens generated   |    5.83 k    |
> | Peak GPU memory (per GPU)  |     44.64 GB    |

---

> > ### Comment · Reviewer_ggJ3 · 2025-08-06
> >
> > I appreciate the authors' thoughtful response and their efforts in conducting additional experiments and efficiency analyses.  The added cross-framework comparisons help to better isolate the impact of the model architecture versus the agentic framework.  Regarding inference efficiency, the authors report detailed metrics such as wall-clock time, model-only latency, token counts, and memory usage.  These additions make the paper more practical and informative, particularly for real-world deployment scenarios.
> >
> > Overall, the rebuttal improves the paper’s clarity and completeness.  I encourage the authors to incorporate these new results and discussions into the final manuscript to further strengthen their contributions.

---

> > > ### Author Response · Authors · 2025-08-06
> > >
> > > Thank you very much for your thoughtful review! We will add these to the paper. Thanks!

---

### Note · Authors · 2025-08-12

We thank the reviewers for the constructive feedback and active engagement. We responded to all the review comments, including the additional comments during the rebuttal discussion. Below, we summarize our responses:

**New experiments**

We added all experiments suggested by reviewers:

**Cross-framework Evaluation.** We validated Co-PatcheR's effectiveness across frameworks, maintaining a 4.2% advantage over SWE-RL-70B when both are tested with Agentless (Reviewer ggJ3).

**Inference efficiency metrics.** We provided a comprehensive efficiency evaluation showing an average end-to-end resolution time of 5.6 minutes per issue with practical GPU memory requirements (Reviewer ggJ3, awx7).

**Separate model for localization and generation.** We validated our joint modeling approach, showing that the unified model achieves better performance (56.20% vs 55.40% pass@20) while maintaining parameter efficiency (Reviewer GmxY, ZdqA).

**All-in-one model comparison.** We compared against a single 14B model handling all tasks, demonstrating that task specialization yields superior performance (43.20% vs 41.40% best@20) (Reviewer ZdqA).


Below, we also summarize the key points in our responses.

**Reviewer ggJ3**

We demonstrated that Co-PatcheR's superior performance comes from the models by testing our models on different frameworks and consistently outperforming baselines.

We provided detailed efficiency metrics to show practical deployment feasibility.


**Reviewer GmxY**

We clarified the distinction between the assertion and non-assertion PoC tests for patch validation.

We demonstrated that combining localization and generation in one model outperforms separate models due to shared reasoning capabilities and task interdependence.

We addressed the inaccurate phrasing issues.

**Reviewer ZdqA**

We demonstrated that combining localization and generation in one model outperforms separate models because of shared reasoning capabilities and task interdependence. However, extending this to an all-in-one model handling all tasks degrades performance because of the lack of PoC diversity.

We clarified the distinction between the assertion and non-assertion PoC tests for patch validation.

**Reviewer awx7**

We clarified the definition of "resolved".

We demonstrated the effectiveness of non-assertion tests through direct comparison when included in the validation framework.

We provided detailed efficiency metrics to show practical deployment feasibility.

---

### Decision · Program_Chairs · 2025-09-17

**Decision:**

Accept (poster)

**Comment:**

The paper received mixed ratings initially. There are a few minor concerns from the reviewers. The rebuttal has addressed most of them, and the reviewers raised their scores. The final decision is acceptance.